# Exposure to conflicts and the continuum of maternal healthcare: Analyses of pooled cross-sectional data for 452,192 women across 49 countries and 82 surveys

**Anu Rammohan**[1], **Astghik Mavisakalyan**[2], **Loan Vu**[1,3], **Srinivas Goli**[4,5]*

**1** Department of Economics, University of Western Australia, Perth, Australia, **2** Astghik Mavisakalyan, Bankwest Curtin Economics Centre, Curtin University, Australia, **3** Vietnam National University of Forestry, Hanoi, Vietnam, **4** Australia India Institute, University of Western Australia, Perth, Australia, **5** Centre for the Study of Regional Development, Jawaharlal Nehru University (JNU), New Delhi, India

* srinivas.goli@uwa.edu.au

**Data Availability Statement:** The authors confirm that all data underlying the findings are fully available without restriction. All data used in this

## Abstract

### Background

Violent conflicts are observed in many parts of the world and have profound impacts on the lives of exposed individuals. The limited evidence available from specific country or region contexts suggest that conflict exposure may reduce health service utilization and have adverse affects on health. This study focused on identifying the association between conflict exposure and continuum of care (CoC) services that are crucial for achieving improvements in reproductive, maternal, newborn, and child health and nutrition (RMNCHN).

### Methods and findings

We combined data from 2 sources, the Demographic Health Surveys (DHS) and the Uppsala Conflict Data Program's (UCDP) Georeferenced Event Dataset, for a sample of 452,192 women across 49 countries observed over the period 1997 to 2018. We utilized 2 consistent measures of conflict—incidence and intensity—and analyzed their association with maternal CoC in 4 key components: (i) at least 1 antenatal care (ANC) visit; (ii) 4 or more ANC visits; (iii) 4 or more ANC visits and institutional delivery; and (iv) 4 or more ANC visits, institutional delivery, and receipt of postnatal care (PNC) either for the mother or the child within 48 hours after birth. To identify the association between conflict exposure and components of CoC, we estimated binary logistic regressions, controlling for a large set of individual and household-level characteristics and year-of-survey and country/province fixed-effects. This empirical setup allows us to draw comparisons among observationally similar women residing in the same locality, thereby mitigating the concerns over unobserved heterogeneity. Around 39.6% (95% CI: 39.5% to 39.7%) of the sample was exposed to some form of violent conflict at the time of their pregnancy during the study period (2003 to 2018). Although access to services decreased for each additional component of CoC in maternal healthcare for all women, the dropout rate was significantly higher among women

study are owned by multiple third-party stakeholders, including Uppasala Conflict Data Program, Uppsala University, and the DHS programme, USAID. The data is made publicly available without restriction at the time of publication from below mentioned public data repositories: • Uppsala Conflict Data Program (UCDP). Uppsala Conflict Data https://ucdp.uu.se/downloads/https://www.pcr.uu.se/research/ucdp/. • DHS M. Demographic and health surveys. Calverton: Measure DHS. 2021 Oct 4. https://dhsprogram.com/data/available-datasets.cfm

**Funding:** The author(s) received no specific funding for this work.

**Competing interests:** The authors have declared that no competing interests exist.

**Abbreviations:** ANC, antenatal care; ASHA, Accredited Social Health Activist; CoC, continuum of care; DHS, Demographic Health Surveys; PNC, postnatal care; RMNCHN, reproductive, maternal, newborn, and child health and nutrition; UCDP, Uppsala Conflict Data Program.

who have been exposed to conflict, relative to those who have not had such exposure. From logistic regression estimates, we observed that relative to those without exposure to conflict, the odds of utilization of each of the components of CoC was lower among those women who were exposed to at least 1 violent conflict. We estimated odds ratios of 0.86 (95% CI: 0.82 to 0.91, $p < 0.001$) for at least 1 ANC; 0.95 (95% CI: 0.91 to 0.98, $p < 0.005$) for 4 or more ANC; and 0.92 (95% CI: 0.89 to 0.96, $p < 0.001$) for 4 or more ANC and institutional delivery. We showed that both the incidence of exposure to conflict as well as its intensity have profound negative implications for CoC. Study limitations include the following: (1) We could not extend the CoC scale beyond PNC due to inconsistent definitions and the lack of availability of data for all 49 countries across time. (2) The measure of conflict intensity used in this study is based on the number of deaths due to the absence of information on other types of conflict-related harms.

## Conclusions

This study showed that conflict exposure is statistically significantly and negatively associated with utilization of maternal CoC services, in each component of the CoC scale. These findings have highlighted the challenges in achieving the Sustainable Development Goal 3 in conflict settings, and the need for more concerted efforts in ensuring CoC, to mitigate its negative implications on maternal and child health.

## Author summary

### Why was this study done?

- The concept of continuum of care (CoC) has emerged as an important guiding principle targeting improvements in reproductive, maternal, newborn, and child health and nutrition (RMNCHN).

- Previous literature on the links between conflicts and CoC has examined the role of socioeconomic and demographic factors on CoC and focused on single country settings at one point in time.

- The influence of conflict exposure on the uptake of CoC services has not been previously analyzed in a multicountry context, and little is known on the links between conflict exposure and CoC utilization using consistent measures of conflict and CoC for a large group of countries over a period of time.

### What did the researchers do and find?

- We combined georeferenced data from 2 major sources for a sample of 452,192 women across 49 countries over the period 1997 to 2018.

- We estimated binary logistic regression models on the link between conflict exposure and 4 components of CoC in maternal health services, controlling for a wide range of

individual- and household-level characteristics as well as year-of-survey and country/
province fixed effects.

- Exposure to conflict—both in terms of incidence and intensity—was found to be negatively associated with CoC across the entire spectrum.

## What do these findings mean?

- This study showed that violent conflict is an important driver of disruptions in CoC in developing countries.

- More policy efforts need to be directed to ensuring CoC in maternal and child health services in conflict-prone parts of the world.

## Introduction

In the last 15 years, the concept of continuum of care (hereafter CoC) has emerged as an important guiding principle targeting improvements in reproductive, maternal, newborn, and child health and nutrition (RMNCHN) [1–4]. In a systematic review of nearly 1,000 studies, Bhutta and colleagues found strong links between reproductive, maternal, newborn, and child health indicators, thus emphasizing the significance of CoC in RMNCHN for the prevention of maternal and childhood mortalities [5]. The CoC concept originally proposed in 1978 by Tanahashi [6] refers to a key package of integrated maternal, newborn, and child health services from pregnancy and delivery to the postnatal period, which are critical for gains in maternal and child survival. It has 2 components—the first is a life cycle approach, which includes adolescence, pregnancy, childbirth, postnatal period, and childhood. The second important component of CoC relates to the location of care and may include (a) household and community care; (b) outpatient and outreach services; and (c) hospital and health facilities. The CoC concept is particularly critical for newborn and maternal health in developing countries. Despite significant progress globally in maternal and newborn child health outcomes, maternal and neonatal mortality remain unacceptably high in many developing countries [7–11].

Recent research has examined the CoC concept to understand maternal and healthcare usage [4,11,12] and the role of trained community health workers in improving the utilization of health services across the continuum [13]. In particular, Yeji and colleagues [12] study found that the CoC completion in Ghana was only 8% in their sample, with the greatest decline being between delivery and postnatal care (PNC) within 48 hours postpartum. Similarly, using nationally representative data from Pakistan, Iqbal and colleagues [4] showed that there was an increase in CoC completion rates from 15% to 27% over the period 2006 to 2012, and CoC was higher among women with better education, autonomy, and socioeconomic backgrounds. Agarwal and colleagues [13] found that exposure to the Accredited Social Health Activist (ASHA) program in India improved the utilization of CoC services.

However, while these studies have examined the role of socioeconomic and demographic characteristics on CoC, to the best of our knowledge, the influence of conflict exposure on uptake of CoC services has not been previously analyzed in a multicountry context. Conflicts

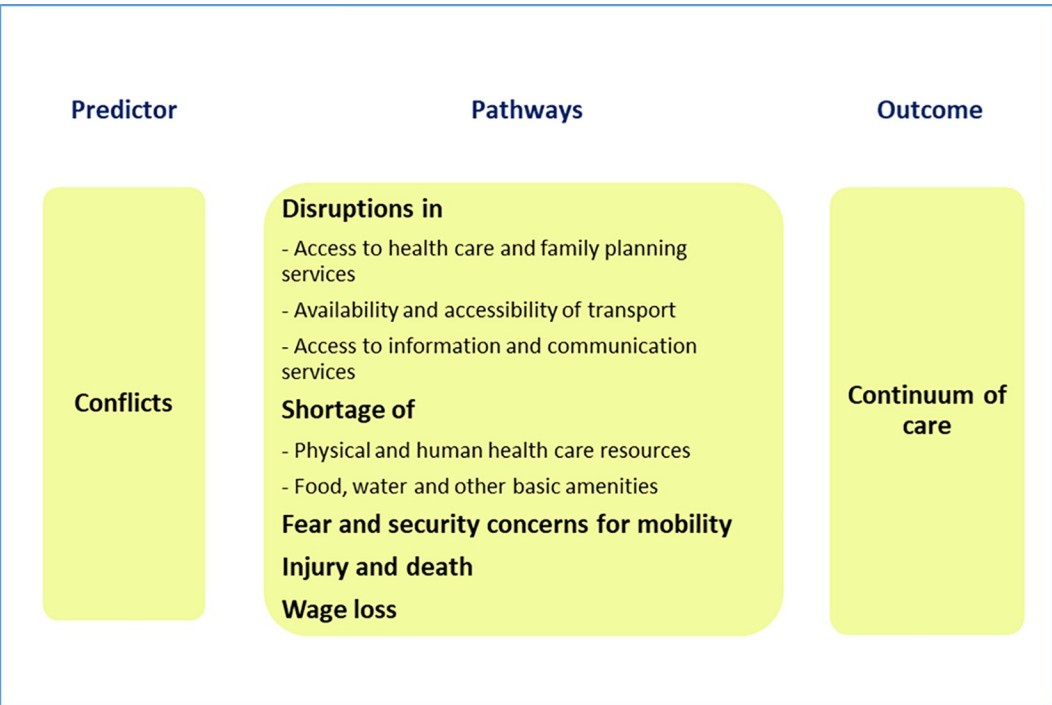

**Fig 1. Logic model showing the influence of conflict on maternal CoC.** CoC, continuum of care.

have the potential to adversely affect the provision of critical maternal and newborn services, both directly and indirectly. Fig 1 presents a Logic model of the potential pathways of the influence of conflicts on CoC. By disrupting health infrastructure, mobility of health workers, and availability and accessibility of healthcare provisions, conflict exposure may negatively affect the utilization of care throughout the continuum. Also, conflict-led fear and security concerns, availability of transport facilities, wage loss, and injury or death of family members or neighbors also influence health-seeking behavior [14–24].

Previous research, in particular from the BRANCH consortium case studies, have demonstrated that conflict impacts negatively on maternal and child health, child birth weight, and immunization rates [15–24]. However, a majority of these studies are country and region specific in their scope, and their findings are context specific. A comprehensive and methodologically robust assessment of the association between conflict and a range of maternal healthcare services, particularly from a CoC perspective, is critically needed. In this paper, we empirically estimated the relationship between exposure to conflict and utilization of care through the continuum in a large sample of developing countries.

## Methods

### Data sources

Our analysis combined data from 2 key sources, conflict data from the Uppsala Conflict Data Program (UCDP) and data on CoC from the Demographic Health Surveys (DHS) [25,26]. Data on conflict were obtained from UCDP's Georeferenced Event Dataset version 19.1, which included information on the dates, locations, and the number of deaths associated with violent conflict around the world in the period from 1989 to 2018 [27].

Data on the components of CoC in maternal healthcare came from the DHS—a collection of nationally representative repeated cross-sectional surveys conducted in over 90 developing

countries since 1984. The DHS interviewed women of childbearing age (15 to 49 years) using a standard questionnaire across all countries and included detailed questions on the socioeconomic and demographic characteristics of surveyed women and their households, the birth histories of all children born in the 5 years before the survey, and information relating to the use of healthcare services. The DHS provides georeferenced information on the residential location of households, including the names and GPS coordinates in more recent survey rounds [26].

We developed a detailed procedure to combine the data on violent conflicts with data on maternal healthcare utilization and other individual- and household-level socioeconomic and demographic characteristics across the spatial and temporal domains. In the spatial domain, we utilized the names of provinces at the first administrative level that are available in both datasets. These are the names of the provinces where a conflict took place in the UCDP data and the names of the provinces of residence in the DHS dataset. If there are differences in the names of provinces across the 2 datasets, we manually reconciled these differences for merging purposes. In total, there are 523 different provinces across the 49 countries in our sample. We did not utilize the data on latitude/longitude positions in the datasets for 2 reasons. Firstly, the GPS data are available for a smaller sample of DHS countries—only 33 out of 49 countries have datasets with these variables. We chose not to compromise on the number of countries represented in the sample, given that the focus of the current study is on providing large multi-country evidence on the link between conflict exposure and its influence on continuum of maternal healthcare. Secondly, to safeguard the respondent's confidentiality, the DHS randomly display the GPS latitude/longitude positions up to 2 kilometers in urban areas and 5 kilometers in rural areas. As a result, we were not able to accurately capture the respondent's location of residence.

In the temporal domain, we utilized information on the start and end year of the conflict in UCDP, alongside information on the month of birth of the last child born in the 5 years prior to the interview date in the DHS to determine the year of their conception. We focused on the last child because complete information on the continuum of maternal healthcare within the DHS was only elicited with reference to the last-born child. We assumed that a child was conceived in the year before their birth year if they were born between January and August, otherwise the year of conception coincides with the year of birth. For each woman in each province, we assigned the information on conflict events that took place from the year of conception of the last child to the year of the interview. The oldest child in our sample was conceived in 1997, and, accordingly, our analysis covers the history of conflict dating back to 1997.

This study is reported as per the Strengthening the Reporting of Observational Studies in Epidemiology (STROBE) guidelines (S1 Table). However, this study did not have a prespecified analysis plan.

## Study population

Our final sample includes 452,192 women aged 15 to 49 years from 82 DHS surveys across 49 countries over the period 2003 to 2018. To the best of our knowledge, our study is unique in measuring both the incidence and intensity of conflict exposure for a large sample of women from heterogeneous geographical and cultural settings. Details on sample selection methodology are presented in Fig 2 and S2 Table. We showed the flow of information through the different phases of the sample selection process using deductive reasoning of inclusion and exclusion criteria for arriving at the final sample of the study. Originally, we accessed data from 137 DHS surveys across 59 countries (excluding India since it accounted for about one-third of the total sample). Fifty DHS surveys were excluded due to a lack of information on the

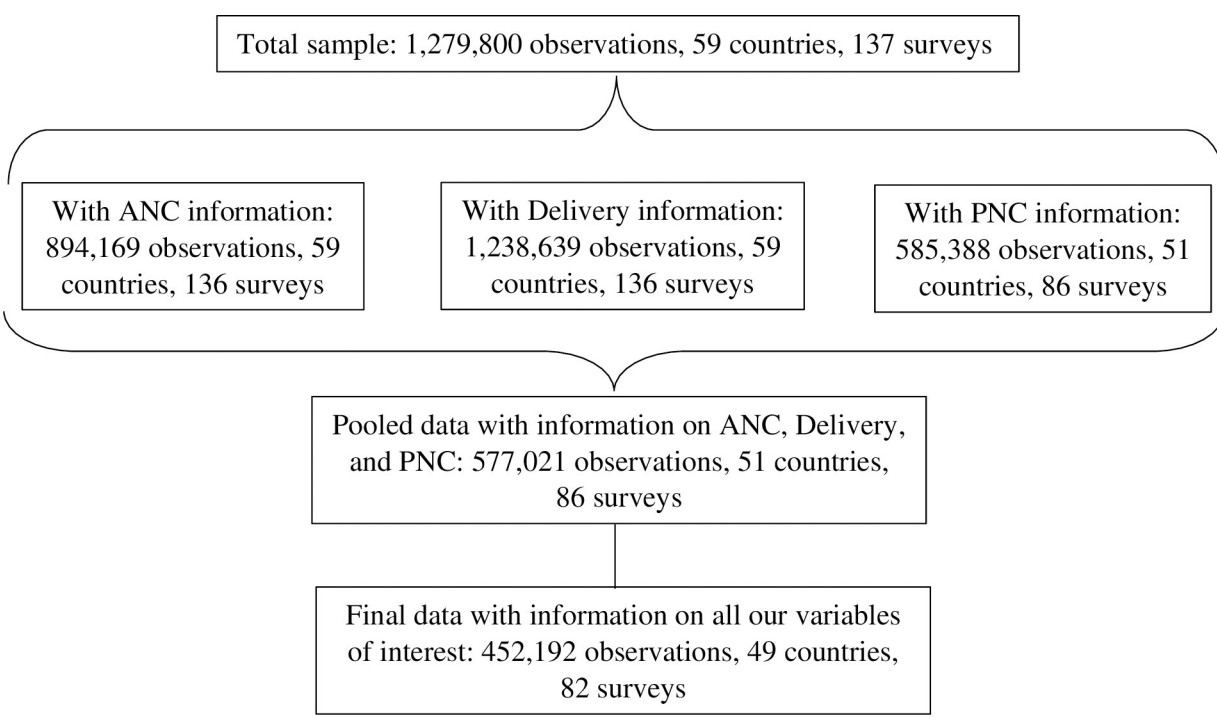

Note: Originally, we could access data from 137 DHS surveys across 59 countries (excluding India since this country accounted for about one-third of the total sample).

There is 1 DHS survey without information on ANC: Jordan (2009).

There are 68 DHS surveys without information on PNC using variable "m50": Albania (2008, 2017), Angola (2015), Armenia (2015), Azerbaijan (2006), Bangladesh (2007), Bolivia (2008), Burundi (2016), Cambodia (2005, 2010), Colombia (2010, 2015), Congo (2005), Congo Democratic (2007), Egypt (2008), Eswatini (2006), Ethiopia (2008), Ghana (2008), Guinea (2018), Guyana (2009), Haiti (2005, 2016), Honduras (2005), Indonesia (2007, 2017), Jordan (2007, 2009, 2017), Kenya (2008), Lesotho (2009), Liberia (2007), Madagascar (2008), Mali (2006, 2018), Myanmar (2015), Namibia (2006), Nepal (2006, 2016), Niger (2006), Nigeria (2008, 2018), Pakistan (2006, 2017), Papua New Guinea (2016), Peru (2003, 2007, 2009, 2010, 2011, 2012), Philippines (2008, 2017), Rwanda (2007), Senegal (2017), Sierra (2008), South Africa (2016), Tajikistan (2017), Tanzania (2010, 2015), Turkey (2003, 2008), Uganda (2006, 2016), Ukraine (2007), Zambia (2007, 2018), Zimbabwe (2005, 2015). To deal with this issue, we use information on PNC from variable "m51a" alternatively collected in 17 DHS surveys: Albania (2008), Bangladesh (2007), Bolivia (2008), Cambodia (2005, 2010), Colombia (2010), Congo Democratic (2007), Eswatini (2006), Guyana (2009), Haiti (2005), Jordan (2007), Lesotho (2009), Liberia (2007), Niger (2006), Nigeria (2008), Pakistan (2006), and Philippines (2008).

We additionally exclude 4 DHS surveys without information on our interest of variables: Colombia (2005, 2010), Pakistan (2006), and Turkey (2013). Finally, we end up with the final sample of 82 DHS surveys across 49 countries.

**Fig 2. Study sample selection process for outcome variables.** ANC, antenatal care; DHS, Demographic Health Surveys; PNC, postnatal care.

various dimensions of continuum of maternal healthcare, whereas 4 further surveys were dropped due to unavailability of other key variables of interest.

The key outcome measure in our analyses is CoC in 4 key components of maternal healthcare. These include (i) at least 1 antenatal care (ANC) visit (ANC-1); (ii) 4 or more ANC visits (ANC-4); (iii) 4 or more ANC visits (ANC-4) and institutional delivery; and (iv) 4 or more ANC visits (ANC-4), institutional delivery, and receipt of PNC either for the mother or the child within 48 hours after birth. The 4 indicators chosen for assessing CoC in maternal care

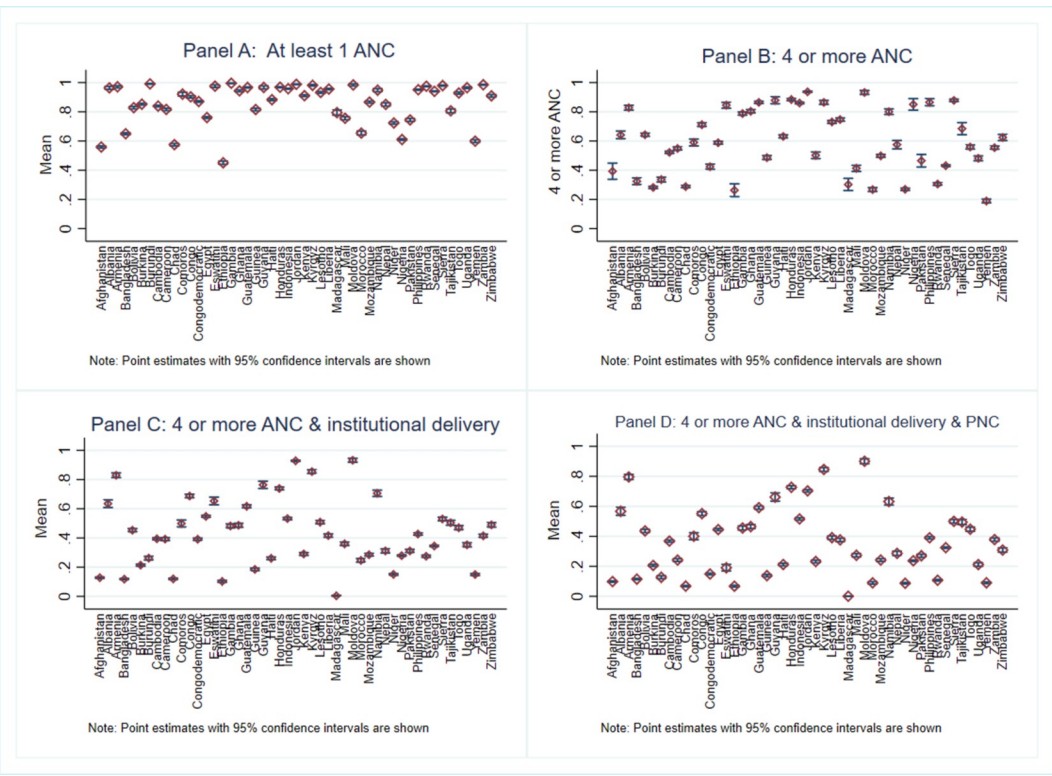

**Fig 3. Levels of different components of CoC in maternal healthcare by countries.** ANC, antenatal care; CoC, continuum of care; PNC, postnatal care.

are part of the World Health Organization's (WHO) recommendation 2 with regard to appropriate health indicators for maternal and children's health [28]. We could not consider the scale of continuum up to child health outcomes due to considerable variation in definition and measurement methods, and lack of availability of information across our sample [26].

The country-level data of the 4 outcome variables across the 49 countries included in our analysis are presented in Fig 3A–3D. There are large differences across countries in the prevalence of CoC in all 4 key components. This observation serves to further validate the focus on understanding the underlying causes of such differences in the current study.

### Explanatory variables

The main explanatory variables in our analysis are measures of exposure to violent conflict during pregnancy in the lifetime pregnancy history of women. We have determined whether women are exposed to conflict events by matching the time of pregnancy from pregnancy history and the time of occurrence of conflict events at the provincial level as described in detail above.

We used information on different types of conflict in UCDP to construct our explanatory variables of interest. UCDP defines a violent event as "an individual incident of lethal violence occurring at a given time and place" [27]. The dataset covered 3 types of violent conflict. They are (i) Type 1, which featured the involvement of government as one of the 2 parties; (ii) Type 2, which involved the use of armed force between groups either of which is the government; and (iii) Type 3, which featured the use of armed force by any party against civilians. The formal definitions of these 3 types of violence are presented in Table 1.

**Table 1. Conflict variables and definitions.**

| Types of violence * categorized by the UCDP | |
|---|---|
| Type 1 (State-based conflict) | The use of armed force between 2 parties, of which at least one is the government of a state |
| Type 2 (Nonstate conflict) | The use of armed force between 2 organized armed groups, neither of which is the government of a state |
| Type 3 (One-sided violence) | The use of armed force by the government of a state or by a formally organized group against civilians |
| **Exposure to at least 1 violent conflict** | |
| Conflict_exposure | = 1 if the child was exposed to at least 1 violent event, 0 otherwise |
| **4 categories of exposure intensity (based on death/year)** | |
| Conflict_intensity | Observations with positive deaths per year between the earliest and the latest violence events are categorized into 3 terciles from 1–3, otherwise 0 |
| Conflict_intensity0 | = 1 if Conflict_intensity = 0, 0 otherwise |
| Conflict_intensity1 | = 1 if Conflict_intensity = 1, 0 otherwise |
| Conflict_intensity2 | = 1 if Conflict_intensity = 2, 0 otherwise |
| Conflict_intensity3 | = 1 if Conflict_intensity = 3, 0 otherwise |

*The term "violence" is originally used by the UCDP Georeferenced Event Dataset. Variables are constructed accounting for conflict within the women's province of residence from conception year to the year of the interview. UCDP, Uppsala Conflict Data Program.

To explore whether conflicts affected CoC in maternal healthcare, we constructed 2 binary measures of conflict exposure taking into account all 3 types of conflict. The first variable, "conflict exposure," takes on a value of 1 if a woman has been exposed to at least 1 violent event of any type since conception of her child to the time she is observed during the interview, 0 otherwise. Based on this measure, nearly 40% of women in the sample were exposed to conflicts (Table 2).

**Table 2. Descriptive statistics of the study variables.**

| | Mean/Proportion | |
|---|---|---|
| | **No exposure to conflict (95% CI)** | **Exposed to at least 1 violent conflict (95% CI)** |
| **Dependent variables** | | |
| At least 1 ANC | 87.6 (87.5–87.7) | 73.2 (73.0–73.4) |
| 4 or more ANC | 56.9 (56.7–57.1) | 44.6 (44.3–44.7) |
| 4 or more ANC and institutional delivery | 41.1 (40.9–41.3) | 29.9 (29.7–30.2) |
| 4 or more ANC and institutional delivery and PNC | 34.4 (34.2–34.6) | 23.9 (23.7–24.1) |
| **Explanatory variables** | | |
| Exposed to at least 1 conflict violence | | 39.6 (39.5–39.7) |
| 4 categories of exposure intensity (based on death/year) | | |
| Conflict_intensity0 | | 62.1 (61.9–62.2) |
| Conflict_intensity1 | | 13.3 (13.2–13.4) |
| Conflict_intensity2 | | 12.9 (12.8–13.0) |
| Conflict_intensity3 | | 11.7 (11.6–11.8) |
| Child is male | 50.9 (51.0–51.5) | 51.3 (51.0–51.5) |
| Child is female | 49.0 (48.8–49.2) | 48.7 (48.5–48.9) |
| Child is part of multiple births | 1.8 (1.7–1.8) | 1.5 (1.4–1.5) |
| Child is not part of multiple births | 98.2 (98.2–98.3) | 98.5 (98.5–98.6) |

(*Continued*)

**Table 2.** (Continued)

| | Mean/Proportion | |
| --- | --- | --- |
| | No exposure to conflict (95% CI) | Exposed to at least 1 violent conflict (95% CI) |
| Child's birth order | | |
| First birth | 20.6 (20.5–20.8) | 20.0 (19.8–20.2) |
| Second birth | 20.6 (20.4–20.7) | 20.1 (19.9–20.3) |
| Third birth+ | 58.8 (58.6–59.0) | 59.9 (59.7–60.1) |
| Mother's age | | |
| 15–19 years | 6.1 (6.1–6.2) | 6.0 (5.9–6.1) |
| 20–29 years | 48.4 (48.2–48.6) | 49.5 (49.3–49.7) |
| 30 years+ | 45.5 (45.3–45.7) | 44.5 (44.3–44.7) |
| Mother's age at first birth | | |
| 15–19 years | 53.2 (53.1–53.4) | 55.0 (54.8–55.3) |
| 20–29 years | 44.4 (44.2–44.6) | 42.6 (42.4–42.8) |
| 30 years+ | 02.4 (02.3–02.4) | 02.4 (02.3–02.4) |
| Mother's age at first cohabitation | 18.9 (18.9–18.9) | 18.5 (18.5–18.5) |
| 15–19 years | 62.0 (61.8–62.2) | 65.8 (65.5–66.0) |
| 20–29 years | 36.1 (35.9–36.3) | 32.6 (32.3–32.8) |
| 30 years+ | 01.9 (01.9–02.0) | 01.7 (01.6–01.7) |
| Mother with no education | 37.7 (37.5–37.9) | 42.8 (42.7–43.1) |
| Mother completed primary education | 31.6 (31.4–31.7) | 23.9 (23.7–24.1) |
| Mother completed secondary education | 24.0 (23.8–24.1) | 25.5 (25.3–25.7) |
| Mother completed higher secondary education | 5.3 (5.2–5.4) | 6.9 (6.7–7.0) |
| Household head's age | 42.1 (42.1–42.1) | 41.5 (41.4–41.6) |
| 13–29 years | 17.5 (17.3–17.6) | 17.1 (16.9–17.3) |
| 30–59 years | 69.1 (68.9–69.3) | 70.7 (70.5–70.9) |
| 60 years+ | 13.4 (13.3–13.6) | 12.2 (12.1–12.4) |
| Household head is male | 83.1 (82.9–83.2) | 87.9 (87.7–88.1) |
| Household head is female | 16.9 (16.8–17.1) | 12.1 (11.9–12.2) |
| Household size | 7.5 (7.5–7.5) | 7.2 (7.2–7.2) |
| 1–3 | 11.2 (11.1–11.3) | 11.3 (11.1–11.4) |
| 4–5 | 29.7 (29.5–29.8) | 29.0 (28.8–29.2) |
| 6+ | 59.2 (59.0–59.3) | 59.7 (59.5–59.9) |
| Rural | 69.1 (68.9–69.3) | 66.8 (66.6–67.1) |
| Urban | 30.9 (30.7–31.1) | 33.1 (32.9–33.4) |
| Poorest | 26.2 (25.9–26.3) | 23.3 (23.0–23.5) |
| Poorer | 22.8 (22.6–22.9) | 20.8 (20.5–20.9) |
| Middle | 20.2 (20.0–20.3) | 19.2 (19.0–19.4) |
| Richer | 17.2 (17.1–17.3) | 18.5 (18.3–18.7) |
| Richest | 13.7 (13.5–13.7) | 18.2 (17.9–18.3) |
| Exposed to media (newspaper, radio, TV): Yes | 78.2 (78.1–78.3) | 71.1 (70.9–71.3) |
| Percentage of households in poorest quintile at the provincial level | 23.5 (23.4–23.5) | 21.7 (21.6–21.8) |
| 0–9% | 18.2 (18.1–18.3) | 28.0 (27.8–28.2) |
| 10–19% | 25.9 (25.7–26.1) | 18.6 (18.4–18.8) |
| 20–29% | 27.1 (27.0–27.3) | 25.2 (25.0–25.4) |
| 30%+ | 28.8 (28.6–28.9) | 28.2 (28.0–28.4) |
| Observations | 273,030 | 179,162 |

CI, confidence interval.

Next, we captured the intensity of conflict by employing a measure of the average number of deaths per year of conflict exposure, named "exposure intensity." This measure is estimated by dividing the total number of deaths that have occurred within the period between the earliest and the latest violent events observed in a woman's pregnancy history by the number of years in that period. It is skewed toward 0 with 62% of the sample not been exposure to a violent event that resulted in positive deaths (Table 2). Thus, it is unlikely that the relationship between this measure and our outcome measures is linear. With this consideration in mind, as a first step, we used categorical measures of conflict intensity defined as a set of 4 dummy variables, distinguishing between women without exposure to a violent event that resulted in positive deaths (omitted category) and those across 3 different terciles of nonzero conflict-led deaths distribution (Conflict_intensity0, Conflict_intensity1, Conflict_intensity2, Conflict_intensity3). Approximately 38% of women in the sample had been exposed to a conflict that resulted in deaths throughout their pregnancy (Table 2).

As a robustness check, we used an alternative measure of conflict exposure intensity, employing the square root of the number of deaths (in thousands) per year of conflict exposure. This approach to defining the conflict exposure intensity measure followed Leone and colleagues [22] and mitigates the common concerns around categorizing continuous variables such as increases of type 1 and type 2 errors.

The other explanatory variables used in our analyses include the key characteristics of women, their children, and the household. These include child characteristics such as the child's sex, birth order, whether the child was part of multiple births, and maternal characteristics including age, age at first birth, age at first cohabitation, and education level. The household's economic status was defined using the wealth index quintiles that were available in the DHS dataset. The wealth index is a composite measure of a household's cumulative living standard. Using a consistent methodology for all countries, the DHS calculated the wealth index using easy-to-collect data on a household's ownership of selected assets, types of water access, and sanitation facilities. The DHS recently introduced methodology making use of household sampling weights stratified by rural/urban on a pooled sample and aggregate them for the national level [29]. Accordingly, the wealth index in all the prior datasets were updated to reflect this. We additionally controlled for household size, rural/urban residence, household head's age and sex, and exposure to mass media (newspaper, radio, or TV). We included variables to capture the geographical heterogeneity within the sample by including the proportion of households in the poorest wealth quintile at the provincial level, as well as country and year-of-survey fixed effects.

## Empirical strategy

We empirically estimated the association between conflict exposure and utilization of CoC. The latent propensity of progression at each stage in the CoC in the 4 maternal healthcare components for a woman $i$ residing in province $p$ in country $c$ and interviewed in year $t$, $CoC^*_{ipct}$, was assumed to depend on her exposure to conflict, $ConflictExposure_{ipct}$, together with (i) a series of child/mother/household controls $Z_{ipct}$ from Table 2; (ii) province-level control $X_{pct}$ for the percentage of households in the poorest quintile; and (iii) dummy variables for country and year-of-survey $K_c$ and $W_t$. Unobserved factors $\varepsilon_{ipct}$ further contribute to the propensity of each stage of the progression in CoC, leading to a latent variable model of the form:

$$CoC^*_{ipct} = \alpha ConflictExposure_{ipct} + Z'_{ipct}\nu + X'_{pct}\gamma + K'_c\tau + W'_t\delta + \varepsilon_{ipct} \qquad (1)$$

Our observed outcome measure, $CoC_{ipct}$, was assumed to relate to the latent propensity through the criterion $CoC_{ipct} = 1\,(CoC^*_{ipct} \geq 0)$, so that the probability of progression in CoC under an assumption of a standard logistic distribution of $\varepsilon_{ipct}$ can be described as a binary

logistic regression model as follows:

$$
\begin{aligned}
&P(CoC_{ipct} = 1 | ConflictExposure_{ipct}, \mathbf{Z}_{ipct}, \mathbf{X}_{ipct}, \mathbf{K}_c, \mathbf{W}_{t,}) = \\
&\exp(\alpha ConflictExposure_{ipct} + \mathbf{Z}'_{ipct}\mathbf{\nu} + \mathbf{X}'_{pct}\mathbf{\gamma} + \mathbf{K}'_c\mathbf{\tau} + \mathbf{W}'_t\boldsymbol{\delta})/[1+ \\
&\exp(\alpha ConflictExposure_{ipct} + \mathbf{Z}'_{ipct}\mathbf{\nu} + \mathbf{X}'_{pct}\mathbf{\gamma} + \mathbf{K}'_c\mathbf{\tau} + \mathbf{W}'_t\boldsymbol{\delta})]
\end{aligned}
\tag{2}
$$

The binary logistic regression model is commonly estimated by the maximum likelihood function. Odds ratios are calculated and used for interpretation.

While our data have multilevel structure, our primary interest was in a single-level relationship rather than in parameter heterogeneity across different levels of analysis. Thus, we did not employ a multilevel modeling approach in the current analysis, but we addressed the clusterization of standard errors in our multilevel data structure by estimating cluster-robust standard errors that adjusted the standard errors for correlation within provinces.The inclusion of country fixed effects $\mathbf{K}_c$ in model 2 implies that our analysis compared the relationship between conflict exposure and progression in CoC for women based in the same country. This eliminated the country-level unobserved heterogeneity entirely. However, there may be unobserved factors not just at the country level but also at the province level confounding the estimation of the influence of conflict exposure on CoC. The inclusion of province-level control $\mathbf{X}_{pct}$ to some extent mitigated some of the unobserved heterogeneity at the province level. To address the issue directly, we estimated a version of model 2, where we replaced country dummy variables with province-level variables $\mathbf{L}_p$ (model 3).

## Results

### Summary statistics

In Table 2, we have reported the summary statistics of the main variables in subsamples, disaggregated by whether or not the women were exposed to a violent conflict event or not. Around 39.6% (95% CI: 39.5% to 39.7%) of the sample was exposed to some form of violent conflict at the time of pregnancy during the study period (2003 to 2018). Such exposure to conflict appears to have significant implications for CoC. Although access to services decreased for each additional component of CoC in maternal healthcare for all women, the dropout rate was significantly higher among women who have been exposed to conflict, relative to those who have not had such exposure (Fig 4).

For instance, access to at least 1 ANC service was high, with 87.6% (95% CI: 87.5% to 87.7%) of women in nonconflict areas receiving at least 1 ANC service. However, it was nearly 14 percentage points lower among women exposed to conflict (73.2%, 95% CI: 73.0% to 73.4%). The utilization rate for up to 4 or more ANC services is under 45% for women exposed to conflict but is nearly 57% among women not exposed to conflict. The large differences between the 2 groups of women persisted during the transition from obtaining 4 or more ANCs to accessing institutional delivery facilities. We observed similar results in the terminal component of CoC in maternal healthcare, where the share of women who obtained at least 4 ANCs along with institutional delivery and PNC was lower for the group exposed to conflict (23.9%, 95% CI: 23.7% to 24.1%), relative to the group with no conflict exposure (34%, 95% CI: 34.2% to 34.6%). Moreover, the differences between these 2 groups was statistically significant for all the components of CoC in maternal healthcare.

In terms of demographic characteristics, the mean age of women in our sample did not vary significantly between those not exposed to conflict (29.29 years; 95% CI: 29.27 to 29.32) and those exposed to conflict (29.15 years; 95% CI: 29.12 to 29.18). The share of women living

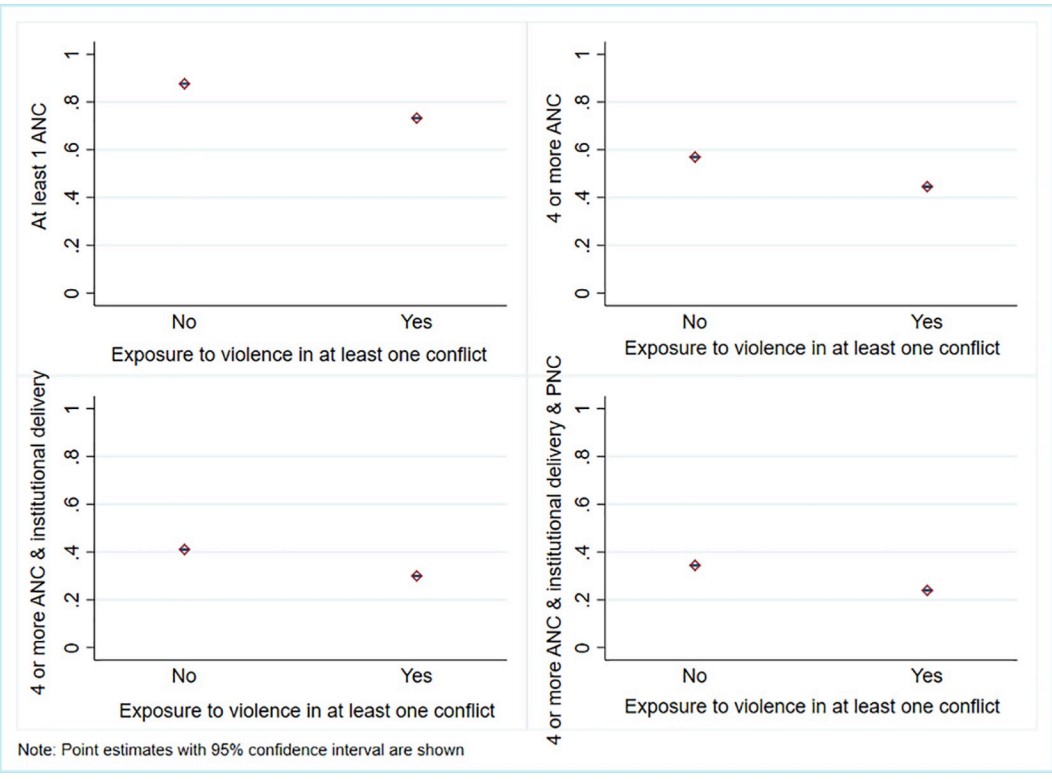

**Fig 4. Levels of different components of CoC in maternal healthcare by the intensity of the conflicts.**

in rural areas is around 69% (95% CI: 68.9% to 69.3%) among those not exposed to conflict relative to 67% (95% CI: 66.6% to 67.1%) among those with conflict exposure.

Notably, a higher proportion of women exposed to conflict are in the highest wealth quintile (18.2%, 95% CI: 17.9% to 18.3%), which is nearly 5 percentage points higher than among women who were not exposed to conflict (13.7%, 95% CI: 13.5 to 13.7). While 42.8% (95% CI: 42.7% to 43.1%) of the women with a history of conflict exposure had no education, the figure is 37.7 (95% CI: 37.5% to 37.9%) among those with no conflict exposure.

## Exposure to conflict and CoC in maternal healthcare

The main empirical results from our analyses are presented in Tables 3–6. We found that exposure to at least 1 violent conflict is statistically significant and negatively associated with utilization of maternal healthcare across the continuum (Table 3). After controlling for an array of socioeconomic and demographic characteristics, we found that relative to those without exposure to conflict, the odds of utilization of each of the components of CoC are lower among those women who were exposed to at least 1 violent conflict. Conflict exposure is negatively associated with components of CoC, with odds ratios of 0.86 (95% CI: 0.82 to 0.91) for at least 1 ANC; 0.95 (95% CI: 0.91 to 0.98) for 4 or more ANC; and 0.92 (95% CI: 0.89 to 0.96) for 4 or more ANC and institutional delivery. However, mere exposure to at least 1 conflict did not show a statistically significant association with utilization of full CoC in maternal healthcare (i.e., utilization of 4 or more ANC visits, institutional delivery, and PNC).

Additionally, we explored whether the association between conflict exposure and utilization of CoC varied by the type of place of conflict. To that end, we reestimated the regressions in Table 3

**Table 3. Associations between exposure to conflict and CoC.**

| | Dependent variable | | | |
|---|---|---|---|---|
| | **At least 1 ANC** | **4 or more ANC** | **4 or more ANC and institutional delivery** | **4 or more ANC and institutional delivery and PNC** |
| | **OR (95% CI) (*p*-value)** | **OR (95% CI) (*p*-value)** | **OR (95% CI) (*p*-value)** | **OR (95% CI) (*p*-value)** |
| Not exposed to any conflict (ref.) | | | | |
| Exposed to at least 1 conflict | 0.86 (0.82, 0.91) (<0.001) | 0.95 (0.91, 0.98) (0.002) | 0.92 (0.89, 0.96) (<0.001) | 0.99 (0.96, 1.04) (0.866) |
| Child is female (ref.) | | | | |
| Child is male | 1.02 (1.00, 1.04) (0.067) | 1.01 (0.99, 1.02) (0.206) | 1.03 (1.02, 1.05) (<0.001) | 1.04 (1.02, 1.05) (<0.001) |
| Child is not part of multiple births (ref.) | | | | |
| Child is part of multiple births | 1.47 (1.36, 1.59) (<0.001) | 1.34 (1.26, 1.42) (<0.001) | 1.75 (1.65, 1.86) (<0.001) | 1.72 (1.62, 1.83) (<0.001) |
| Child's birth order: first birth (ref.) | | | | |
| Child's birth order: second birth | 0.73 (0.71, 0.76) (<0.001) | 0.76 (0.74, 0.78) (<0.001) | 0.65 (0.63, 0.67) (<0.001) | 0.71 (0.68, 0.73) (<0.001) |
| Child's birth order: third birth+ | 0.64 (0.62, 0.66) (<0.001) | 0.66 (0.64, 0.68) (<0.001) | 0.52 (0.50, 0.54) (<0.001) | 0.57 (0.55, 0.59) (<0.001) |
| Mother's age: 15–19 (ref.) | | | | |
| Mother's age: 20–29 | 1.27 (1.22, 1.33) (<0.001) | 1.33 (1.28, 1.38) (<0.001) | 1.28 (1.23, 1.33) (<0.001) | 1.25 (1.20, 1.30) (<0.001) |
| Mother's age: 30+ | 1.26 (1.19, 1.33) (<0.001) | 1.49 (1.43, 1.55) (<0.001) | 1.50 (1.43, 1.56) (<0.001) | 1.49 (1.42, 1.56) (<0.001) |
| Mother's age at first birth: 15–19 (ref.) | | | | |
| Mother's age at first birth: 20–29 | 1.03 (1.00, 1.06) (0.043) | 1.05 (1.02, 1.07) (<0.001) | 1.08 (1.06, 1.11) (<0.001) | 1.09 (1.07, 1.12) (<0.001) |
| Mother's age at first birth: 30+ | 1.01 (0.91, 1.12) (0.879) | 1.15 (1.07, 1.23) (<0.001) | 1.28 (1.20, 1.38) (<0.001) | 1.35 (1.26, 1.45) (<0.001) |
| Mother's age at first cohabitation: 15–19 (ref.) | | | | |
| Mother's age at first cohabitation: 20–29 | 1.18 (1.14, 1.21) (<0.001) | 1.07 (1.05, 1.10) (<0.001) | 1.11 (1.09, 1.14) (<0.001) | 1.08 (1.06, 1.11) (<0.001) |
| Mother's age at first cohabitation: 30+ | 1.16 (1.03, 1.30) (0.012) | 1.02 (0.95, 1.10) (0.591) | 1.12 (1.04, 1.21) (0.004) | 1.04 (0.96, 1.12) (0.378) |
| Mother with no education (ref.) | | | | |
| Mother completed primary education | 2.24 (2.16, 2.31) (<0.001) | 1.57 (1.54, 1.61) (<0.001) | 1.60 (1.56, 1.64) (<0.001) | 1.55 (1.51, 1.60) (<0.001) |
| Mother completed secondary education | 3.75 (3.59, 3.92) (<0.001) | 2.27 (2.21, 2.34) (<0.001) | 2.49 (2.42, 2.57) (<0.001) | 2.31 (2.24, 2.39) (<0.001) |
| Mother completed higher secondary education | 8.29 (7.43, 9.26) (<0.001) | 4.38 (4.15, 4.61) (<0.001) | 5.38 (5.12, 5.67) (<0.001) | 4.29 (4.09, 4.50) (<0.001) |
| Household head's age: 13–29 (ref.) | | | | |
| Household head's age: 30–59 | 1.04 (1.01, 1.08) (0.016) | 1.07 (1.05, 1.10) (<0.001) | 1.11 (1.08, 1.13) (<0.001) | 1.10 (1.07, 1.13) (<0.001) |
| Household head's age: 60+ | 1.07 (1.02, 1.12) (0.002) | 1.10 (1.07, 1.13) (<0.001) | 1.14 (1.10, 1.18) (<0.001) | 1.14 (1.10, 1.18) (<0.001) |
| Household head is female (ref.) | | | | |
| Household head is male | 0.96 (0.93, 1.00) (0.040) | 0.97 (0.95, 0.99) (0.009) | 0.94 (0.92, 0.97) (<0.001) | 0.94 (0.92, 0.97) (<0.001) |
| Household size: 1–3 (ref.) | | | | |
| Household size: 4–5 | 1.03 (1.00, 1.07) (0.113) | 0.99 (0.96, 1.02) (0.559) | 1.01 (0.98, 1.04) (0.347) | 1.02 (0.98, 1.05) (0.330) |

(*Continued*)

**Table 3.** (Continued)

| | Dependent variable | | | |
|---|---|---|---|---|
| | **At least 1 ANC** | **4 or more ANC** | **4 or more ANC and institutional delivery** | **4 or more ANC and institutional delivery and PNC** |
| | **OR (95% CI) (p-value)** | **OR (95% CI) (p-value)** | **OR (95% CI) (p-value)** | **OR (95% CI) (p-value)** |
| Household size: 6+ | 0.94(0.90, 0.98) (0.002) | 0.86 (0.84, 0.89) (<0.001) | 0.87 (0.84, 0.89) (<0.001) | 0.88 (0.85, 0.91) (<0.001) |
| Urban (ref.) | | | | |
| Rural | 0.70 (0.67, 0.73) (<0.001) | 0.84 (0.82, 0.87) (<0.001) | 0.68 (0.66, 0.70) (<0.001) | 0.74 (0.71, 0.76) (<0.001) |
| Household wealth quintile: Poorest (ref.) | | | | |
| Poorer | 1.30 (1.26, 1.34) (<0.001) | 1.23 (1.20, 1.26) (<0.001) | 1.42 (1.38, 1.46) (<0.001) | 1.42 (1.38, 1.46) (<0.001) |
| Middle | 1.69 (1.62, 1.75) (<0.001) | 1.49 (1.45, 1.53) (<0.001) | 1.81 (1.75, 1.86) (<0.001) | 1.83 (1.77, 1.89) (<0.001) |
| Richer | 2.10 (2.01, 2.20) (<0.001) | 1.85 (1.79, 1.91) (<0.001) | 2.44 (2.36, 2.53) (<0.001) | 2.49 (2.40, 2.57) (<0.001) |
| Richest | 3.54 (3.31, 3.78) (<0.001) | 2.74 (2.63, 2.85) (<0.001) | 3.95 (3.79, 4.12) (<0.001) | 3.98 (3.81, 4.16) (<0.001) |
| Did not expose to media (ref.) | | | | |
| Exposed to media (newspaper, radio, TV) | 1.64 (1.59, 1.70) (<0.001) | 1.37 (1.34, 1.40) (<0.001) | 1.38 (1.34, 1.42) (<0.001) | 1.40 (1.36, 1.44) (<0.001) |
| Percentage of households in poorest quintile at the provincial level: 0%–9% (ref.) | | | | |
| Percentage of households in poorest quintile at the provincial level: 10%–19% | 1.10 (1.03, 1.17) (0.007) | 0.96 (0.93, 0.99) (0.041) | 0.83 (0.80, 0.87) (<0.001) | 0.89 (0.86, 0.93) (<0.001) |
| Percentage of households in poorest quintile at the provincial level: 20%–29% | 1.18 (1.11, 1.25) (<0.001) | 0.98 (0.94, 1.01) (0.198) | 0.84 (0.81, 0.87) (<0.001) | 0.89 (0.86, 0.93) (<0.001) |
| Percentage of households in poorest quintile at the provincial level: 30%+ | 0.87 (0.82, 0.92) (<0.001) | 0.85(0.82, 0.88) (<0.001) | 0.67 (0.64, 0.70) (<0.001) | 0.71 (0.69, 0.74) (<0.001) |
| Year dummies | Yes | Yes | Yes | Yes |
| Country dummies | Yes | Yes | Yes | Yes |

ANC, antenatal care; CI, confidence interval; CoC, continuum of care; OR, odds ratio; PNC, postnatal care.

Standard errors are clustered at the district level.

in separate subsamples of rural and urban localities (Table 4). In the rural subsample, the odds of utilization of each of the components of CoC were lower among women with a history of conflict exposure. We found similar results for the relationship between conflict exposure and at least 1 ANC; 4 or more ANC; and 4 or more ANC and institutional delivery in the urban subsample. However, our results showed that conflict exposure was positively associated with utilization of 4 or more ANC visits, institutional delivery, and PNC in the urban subsample.

In the next stage, we investigated the association between the intensity of exposure to conflict and the CoC in maternal healthcare. In Table 5, we present the results of the empirical analysis with an alternative measure of conflict composed of a set of categorical variables corresponding to 3 different categories of intensity of conflict. In this specification, we have included province fixed-effects, thereby drawing comparisons between women based in the same province of a country. This specification, therefore, can be also seen as a robustness check for controlling unobserved province-level heterogeneity.

From the findings in Table 5, we similarly observed a statistically significant and negative association between conflict intensity and utilization of each of the components of CoC and

Table 4. Associations between exposure to conflict and CoC for rural and urban subsamples.

| | Dependent variable | | | |
|---|---|---|---|---|
| | At least 1 ANC | 4 or more ANC | 4 or more ANC and institutional delivery | 4 or more ANC and institutional delivery and PNC |
| | OR (95% CI) (*p*-value) | OR (95% CI) (*p*-value) | OR (95% CI) (*p*-value) | OR (95% CI) (*p*-value) |
| Not exposed to any conflict (ref.) | | | | |
| **Panel A: Rural subsample (N = 308,429 observations)** | | | | |
| Exposed to at least 1 conflict | 0.87 (0.82, 0.93) (<0.001) | 0.96 (0.91, 1.00) (0.056) | 0.92 (0.88, 0.97) (0.003) | 0.96 (0.91, 1.02) (0.174) |
| **Panel B: Urban subsample (N = 143,763 observations)** | | | | |
| Exposed to at least 1 conflict | 0.82 (0.75, 0.91) (<0.001) | 0.98 (0.93, 1.03) (0.367) | 0.97 (0.92, 1.02) (0.261) | 1.09 (1.03, 1.14) (0.001) |
| Individual and household characteristics | Yes | Yes | Yes | Yes |
| Year dummies | Yes | Yes | Yes | Yes |
| Country dummies | Yes | Yes | Yes | Yes |

ANC, antenatal care; CI, confidence interval; CoC, continuum of care; OR, odds ratio; PNC, postnatal care.

Standard errors are clustered at the district level.

the full CoC in maternal healthcare. At each stage of the CoC in maternal healthcare, compared to women who were not exposed to conflict associated with loss of human life, we found that the odds of utilization of CoC up to institutional delivery were significantly lower among those women exposed to conflicts of different levels of intensity: conflict_intensity1 (OR = 0.77, 95% CI: 0.72 to 0.82) and conflict_intensity2 (OR = 0.89, 95% CI: 0.81 to 0.96). Similar results were also observed at the stage of full CoC in maternal healthcare. Relative to women who were not exposed to conflict associated with loss of human life, the odds of utilization of 4 or more ANCs along with institutional delivery and PNC were significantly lower among women exposed to different levels of conflict: conflict_intensity1 (OR = 0.89, 95% CI: 0.84 to 0.95) and conflict_intensity2 (OR = 0.82, 95% CI: 0.76 to 0.87). With the exception of receipt of at least 1 ANC (OR = 0.84, 95% CI: 0.76 to 0.93) and CoC up to institutional delivery (OR = 0.89, 95% CI: 0.81 to 0.91), the greater intensity of conflicts captured through conflict_intensity3 is not statistically significant. However, the estimated odds ratios were less than 1, indicating a lower probability of utilization of CoC in maternal healthcare for women at this level of higher intensity of conflict exposure.

In addition to using categorical variables to capture conflict intensity, we also employed continuous measure of conflict intensity defined as the square root of the number of deaths (in thousand) per year of conflict exposure as a robustness check reported in Table 6. This analysis yielded qualitatively similar results confirming the statistically significant and negative associations between the intensity of exposure to conflict and CoC in maternal healthcare outcomes.

## Socioeconomic factors and CoC in maternal healthcare

The association between other socioeconomic factors and CoC in maternal healthcare are presented in Tables 3 and 4. As expected, utilization of CoC increased monotonically with household wealth, with women from the richest wealth quintile having a higher odds of utilizing the full CoC, relative to those from the poorest wealth quintile.

**Table 5. Associations between intensity of exposure to conflict and CoC.**

| | At least 1 ANC | 4 or more ANC | 4 or more ANC and institutional delivery | 4 or more ANC and institutional delivery and PNC |
|---|---|---|---|---|
| | OR (95% CI) (p-value) | OR (95% CI) (p-value) | OR (95% CI) (p-value) | OR (95% CI) (p-value) |
| Conflict_intensity0 = 1 (ref.) | | | | |
| Conflict_intensity1 | 0.79 (0.73, 0.85) (<0.001) | 0.93 (0.88, 0.98) (0.007) | 0.99 (0.93, 1.04) (0.657) | 0.89 (0.84, 0.95) (<0.001) |
| Conflict_intensity2 | 0.82 (0.76, 0.89) (<0.001) | 0.86 (0.81, 0.90) (<0.001) | 0.77 (0.72, 0.82) (<0.001) | 0.82 (0.76, 0.87) (<0.001) |
| Conflict_intensity3 | 0.84 (0.76, 0.93) (0.001) | 0.98 (0.91, 1.05) (0.547) | 0.89 (0.81, 0.96) (0.004) | 0.96 (0.87, 1.06) (0.405) |
| Child is female (ref.) | | | | |
| Child is male | 1.02 (1.00, 1.04) (0.046) | 1.01 (0.99, 1.02) (0.222) | 1.04 (1.02, 1.05) (<0.001) | 1.04 (1.02, 1.06) (<0.001) |
| Child is not part of multiple births (ref.) | | | | |
| Child is part of multiple births | 1.46 (1.35, 1.58) (<0.001) | 1.32 (1.25, 1.40) (<0.001) | 1.74 (1.64, 1.85) (<0.001) | 1.71 (1.61, 1.82) (<0.001) |
| Child's birth order: first birth (ref.) | | | | |
| Child's birth order: second birth | 0.72 (0.70, 0.75) (<0.001) | 0.76 (0.72, 0.78) (<0.001) | 0.65 (0.63, 0.67) (<0.001) | 0.71 (0.69, 0.73) (<0.001) |
| Child's birth order: third birth+ | 0.63 (0.61, 0.65) (<0.001) | 0.66 (0.64, 0.68) (<0.001) | 0.52 (0.50, 0.53) (<0.001) | 0.57 (0.55, 0.58) (<0.001) |
| Mother's age: 15–19 (ref.) | | | | |
| Mother's age: 20–29 | 1.25 (1.20, 1.31) (<0.001) | 1.30 (1.25, 1.34) (<0.001) | 1.25 (1.20, 1.30) (<0.001) | 1.23 (1.18, 1.28) (<0.001) |
| Mother's age: 30+ | 1.19 (1.13, 1.26) (<0.001) | 1.41 (1.36, 1.47) (<0.001) | 1.43 (1.37, 1.49) (<0.001) | 1.43 (1.37, 1.50) (<0.001) |
| Mother's age at first birth: 15–19 (ref.) | | | | |
| Mother's age at first birth: 20–29 | 1.03 (1.00, 1.06) (0.026) | 1.04 (1.02, 1.06) (<0.001) | 1.08 (1.05, 1.10) (<0.001) | 1.09 (1.06, 1.11) (<0.001) |
| Mother's age at first birth: 30+ | 0.98 (0.89, 1.09) (0.759) | 1.14 (1.06, 1.22) (<0.001) | 1.28 (1.19, 1.37) (<0.001) | 1.35 (1.26, 1.45) (<0.001) |
| Mother's age at first cohabitation: 15–19 (ref.) | | | | |
| Mother's age at first cohabitation: 20–29 | 1.11 (1.08, 1.15) (<0.001) | 1.03 (1.02, 1.06) (0.004) | 1.07 (1.04, 1.09) (<0.001) | 1.04 (1.01, 1.06) (0.003) |
| Mother's age at first cohabitation: 30+ | 1.10 (0.98, 1.24) (0.104) | 0.99 (0.92, 1.07) (0.824) | 1.07 (0.99, 1.16) (0.079) | 0.99 (0.92, 1.07) (0.897) |
| Mother with no education (ref.) | | | | |
| Mother completed primary education | 1.92 (1.86, 1.99) (<0.001) | 1.34 (1.31, 1.37) (<0.001) | 1.37 (1.33, 1.41) (<0.001) | 1.32 (1.28, 1.36) (<0.001) |
| Mother completed secondary education | 3.18 (3.05, 3.32) (<0.001) | 1.98 (1.92, 2.03) (<0.001) | 2.16 (2.10, 2.23) (<0.001) | 2.02 (1.96, 2.09) (<0.001) |
| Mother completed higher secondary education | 6.96 (6.22, 7.78) (<0.001) | 3.80 (3.60, 4.00) (<0.001) | 4.81 (4.57, 5.06) (<0.001) | 3.79 (3.62, 3.98) (<0.001) |
| Household head's age: 13–29 (ref.) | | | | |
| Household head's age: 30–59 | 1.07 (1.04, 1.11) (<0.001) | 1.09 (1.07, 1.12) (<0.001) | 1.12 (1.09, 1.15) (<0.001) | 1.11 (1.09, 1.14) (<0.001) |
| Household head's age: 60+ | 1.07 (1.02, 1.11) (0.004) | 1.11 (1.07, 1.14) (<0.001) | 1.14 (1.10, 1.18) (<0.001) | 1.15 (1.11, 1.19) (<0.001) |
| Household head is female (ref.) | | | | |

(*Continued*)

**Table 5.** (Continued)

| | Dependent variable | | | |
|---|---|---|---|---|
| | **At least 1 ANC** | **4 or more ANC** | **4 or more ANC and institutional delivery** | **4 or more ANC and institutional delivery and PNC** |
| | **OR (95% CI) (p-value)** | **OR (95% CI) (p-value)** | **OR (95% CI) (p-value)** | **OR (95% CI) (p-value)** |
| Household head is male | 0.96 (0.93, 1.00) (0.042) | 0.96 (0.94, 0.98) (<0.001) | 0.96 (0.94, 0.98) (<0.001) | 0.95 (0.92, 0.97) (<0.001) |
| Household size: 1–3 (ref.) | | | | |
| Household size: 4–5 | 1.01 (0.97, 1.05) (0.626) | 0.99 (0.96, 1.02) (0.425) | 1.01 (0.98, 1.04) (0.655) | 1.01 (0.98, 1.05) (0.411) |
| Household size: 6+ | 0.93 (0.89, 0.97) (0.001) | 0.88 (0.85, 0.90) (<0.001) | 0.88 (0.85, 0.91) (<0.001) | 0.90 (0.87, 0.93) (<0.001) |
| Urban (ref.) | | | | |
| Rural | 0.70 (0.67, 0.74) (<0.001) | 0.84 (0.81, 0.86) (<0.001) | 0.67 (0.65, 0.69) (<0.001) | 0.71 (0.69, 0.74) (<0.001) |
| Household wealth quintile: Poorest (ref.) | | | | |
| Poorer | 1.37 (1.32, 1.41) (<0.001) | 1.27 (1.24, 1.30) (<0.001) | 1.48 (1.44, 1.52) (<0.001) | 1.48 (1.43, 1.52) (<0.001) |
| Middle | 1.82 (1.75, 1.89) (<0.001) | 1.55 (1.50, 1.59) (<0.001) | 1.93 (1.87, 2.00) (<0.001) | 1.92 (1.86, 1.99) (<0.001) |
| Richer | 2.38 (2.27, 2.49) (<0.001) | 1.95 (1.89, 2.01) (<0.001) | 2.69 (2.59, 2.78) (<0.001) | 2.64 (2.55, 2.74) (<0.001) |
| Richest | 4.01 (3.76, 4.28) (<0.001) | 2.91 (2.79, 3.02) (<0.001) | 4.44 (4.25, 4.64) (<0.001) | 4.28 (4.10, 4.48) (<0.001) |
| Did not expose to media (ref.) | | | | |
| Exposed to media (newspaper, radio, TV) | 1.56 (1.52, 1.61) (<0.001) | 1.33 (1.30, 1.36) (<0.001) | 1.31 (1.28, 1.35) (<0.001) | 1.34 (1.31, 1.38) (<0.001) |
| Percentage of households in poorest quintile at the provincial level: 0%–9% (ref.) | | | | |
| Percentage of households in poorest quintile at the provincial level: 10%–19% | 1.65 (1.44, 1.88) (<0.001) | 1.20 (1.13, 1.29) (<0.001) | 0.99 (0.91, 1.07) (0.786) | 1.06 (0.97, 1.15) (0.202) |
| Percentage of households in poorest quintile at the provincial level: 20%–29% | 1.59 (1.38, 1.83) (<0.001) | 1.25 (1.15, 1.35) (<0.001) | 1.15 (1.05, 1.25) (0.002) | 1.18 (1.08, 1.30) (<0.001) |
| Percentage of households in poorest quintile at the provincial level: 30%+ | 1.12 (0.97, 1.29) (0.126) | 1.14 (1.05, 1.24) (0.002) | 1.10 (1.00, 1.21) (0.045) | 1.06 (0.96, 1.18) (0.270) |
| Year dummies | Yes | Yes | Yes | Yes |
| Province dummies | Yes | Yes | Yes | Yes |

ANC, antenatal care; CI, confidence interval; CoC, continuum of care; OR, odds ratio; PNC, postnatal care.

Standard errors are clustered at the district level.

In terms of mother's characteristics, better maternal education was associated with a higher probability of utilizing the full CoC. Relative to women with no education, a mother with completed primary education had higher odds of utilizing the full CoC (OR = 1.55, 95% CI: 1.51 to 1.60), increasing for mothers with a secondary level of education (OR = 2.31, 95% CI: 2.24 to 2.39) and for mothers with higher than a secondary level of education (OR = 4.29, 95% CI: 4.09 to 4.50). Exposure to mass media (such as newspapers, radio, and TV) was also associated with higher odds of utilization of full CoC in maternal healthcare (OR = 1.40, 95% CI: 1.36 to 1.44). A child's birth order and rural residence were negatively associated with the utilization of CoC. A male child had a higher probability of receiving full CoC than female children. We had similar results in alternative model estimations reported in Table 4.

**Table 6. Associations between intensity of exposure to conflict and CoC using continuous measure of conflict intensity.**

| | Dependent variable | | | |
|---|---|---|---|---|
| | **At least 1 ANC** | **4 or more ANC** | **4 or more ANC and institutional delivery** | **4 or more ANC and institutional delivery and PNC** |
| | **Coefficient (95% CI) (*p*-value)** | **Coefficient (95% CI) (*p*-value)** | **Coefficient (95% CI) (*p*-value)** | **Coefficient (95% CI) (*p*-value)** |
| Square root of thousands death per year of conflict exposure | −0.088 (−0.25, 0.07) 0.281) | −0.043 (−0.14, 0.06) (0.408) | −0.258 (−0.38, −0.14) ($<$0.000) | −0.120 (−0.26, 0.01) (0.084) |
| Individual and household characteristics | Yes | Yes | Yes | Yes |
| Year dummies | Yes | Yes | Yes | Yes |
| Province dummies | Yes | Yes | Yes | Yes |

ANC, antenatal care; CI, confidence interval; CoC, continuum of care; PNC, postnatal care.

Standard errors are clustered at the district level.

## Discussion

To the best of our knowledge, CoC studies have to date not assessed the association between exposure to conflict and maternity healthcare utilization. Using a multicountry sample of women over a 15-year period, we found that conflict exposure broke the continuum of maternal healthcare across the scale of the continuum. We showed that both the incidence of exposure to conflict as well as its intensity were negatively associated with the CoC. Using both categorical and continuous measures of conflict intensity based on the number of battle deaths, we were able to demonstrate that higher intensity of conflict exposure was associated with a lower likelihood of CoC in maternal healthcare. These findings hold even after controlling for all other observable socioeconomic, demographic, and regional characteristics of women. Our robustness checks using subsamples from rural areas also broadly supported the main findings of the study. However, conflict exposure was, in fact, positively associated with utilization of 4 or more ANC visits, institutional delivery, and PNC in the urban subsample. What this possibly suggests is that the care deficiencies at earlier stages of CoC due to conflict exposure could have led to complications, triggering increases in PNC utilization. We observe this only in the urban subsample where there is a wider availability of infrastructure to accommodate such needs in urban localities.

Although findings across the studies are strictly not comparable due to geographical context specificity, study design, differences in outcome variable selected, and statistical tools applied, our findings are qualitatively similar to previous country-specific and regional-level studies that examined the links between exposure to conflict and individual components of maternal healthcare indicators [15–24,30–37]. Therefore, our conclusions provide validation for previous country- or region-specific case studies that pointed to the links between exposure to conflict and individual components of maternal healthcare indicators [15–24,30–37].

### Strengths and limitations

Our research is unique in the context of the literature on the relationship between exposure to conflicts and maternal health that has so far been limited to specific country or region contexts [15–24,30–36]. The main strength of our study is our ability to draw robust conclusions on the links between conflict exposure and CoC using a relatively large sample of 452,192 women across 49 countries over 20 years. To the best of our knowledge, our study represents one of the first systematic efforts to provide evidence on the links between violent conflict and CoC

in maternal health. We have examined full implications of conflict on maternal healthcare on the continuum scale rather than assessing its associations with individual components of maternal healthcare. This is an important focus, given that conflict exposure breaks the continuity of maternal healthcare across the scale of the continuum.

Furthermore, using an innovative research design that drew comparisons among observationally similar women residing within the same country or province setting, we were able to address important sources of unobserved heterogeneity, thereby isolating the influence of conflict from other potential confounding factors such as institutions, geography, and culture.

However, our study suffers from a few limitations. (1) We could not extend the CoC scale beyond PNC due to inconsistent definitions and a lack of availability of data for all 49 countries across time. (2) The UCDP uses a consistent measure of conflict for all countries and is generally considered to be reliable. However, we acknowledge that there is potential for measurement errors in large multicountry studies. Moreover, the lack of availability of different types of conflict-related damages limits us to use only deaths as a proxy of conflict intensity. (3) Battle deaths alone do not determine the intensity of the conflicts; other aspects such as the duration of the conflict and related disturbances (e.g., damage to health infrastructure and injuries to health workers need to be considered while constructing conflict severity in a geographical setting). However, in the absence of this information, battle deaths provide the best proxy for the gravity of the conflict. (4) While our study has provided convincing evidence on the link between conflict exposure and CoC, we were unable to empirically identify the mechanism underlying this link. It is likely that conflict exposure affects utilization of care throughout the continuum through disruptions to health infrastructure, health workers' mobility, healthcare provisions, and changes in health-seeking behavior. Identifying the precise mechanisms through which conflict might affect healthcare utilization is an important direction for future research. Furthermore, the analyses in this paper were based on commonly available measures of conflict for all countries. It is possible that our conflict variables may have some unobservable reporting errors. Moreover, in order to provide a holistic perspective on conflict-led damages to healthcare delivery and infrastructure, future studies should also provide some qualitative assessment of the impact of conflict.

Summing up, despite potential data-related limitations, our study has significantly contributed to the existing literature in 3 ways: (1) We found robust evidence on the relationship between CoC in maternal healthcare and conflict exposure of women using a large sample from diverse geographical and cultural settings. (2) Despite limited available information, our conflict measures are unique and our results are in keeping with previous single country studies. (3) Our methodology has controlled for unobservable heterogeneity and omitted variable bias across countries and provinces using country and provincial fixed-effects over time.

## Conclusions

In conclusion, we showed that exposure to at least 1 violent conflict event or intensity of conflicts are negatively associated with utilization of maternal healthcare across the continuum. From a policy perspective, our findings suggest that through its adverse influence on maternal and child healthcare utilization, violent conflicts pose a big challenge for achieving the Sustainable Development Goal 3 [30–35]. Moreover, they have the potential to undo the global progress achieved in improving maternal healthcare. Thus, aside from efforts to foster socioeconomic development, countries need to prioritize peace and harmony to achieve improvements in CoC in maternal healthcare. In conflict settings, in order to ensure CoC, concerted efforts need to be directed to mitigate the negative implications of conflict on maternal and child health. More broadly, as suggested in previous studies [37–40], there is a greater

need for humanitarian actors to navigate and negotiate the impediments to maternal health-care delivery. In particular, from a practical perspective, it is critical that healthcare resources are strengthened to develop resilience to potential conflict-related disruptions. Access to and use of quality healthcare can be ensured through developing capacity to mitigate, adapt, and recover from conflict-related vulnerabilities. Also, there is need for conflict, country, community, and cultural-specific absorptive, adaptive, and transformative capacities to overcome shocks, and stresses for women, their newborns, and other children and families that can affect healthcare in fragile settings.

## Supporting information

**S1 Table. STROBE Checklist.** STROBE, Strengthening the Reporting of Observational Studies in Epidemiology.
(DOCX)

**S2 Table. Number of observations by information availability.**
(DOCX)

**S3 Table. Conflict status during 1997 to 2018 in 49 countries in the final sample.**
(DOCX)

## Author Contributions

**Conceptualization:** Anu Rammohan, Astghik Mavisakalyan, Srinivas Goli.

**Data curation:** Loan Vu, Srinivas Goli.

**Formal analysis:** Loan Vu, Srinivas Goli.

**Investigation:** Anu Rammohan, Astghik Mavisakalyan, Loan Vu, Srinivas Goli.

**Methodology:** Anu Rammohan, Astghik Mavisakalyan, Loan Vu, Srinivas Goli.

**Project administration:** Anu Rammohan, Srinivas Goli.

**Resources:** Anu Rammohan, Astghik Mavisakalyan, Srinivas Goli.

**Software:** Loan Vu, Srinivas Goli.

**Supervision:** Anu Rammohan, Astghik Mavisakalyan, Srinivas Goli.

**Validation:** Anu Rammohan.

**Visualization:** Srinivas Goli.

**Writing – original draft:** Anu Rammohan, Astghik Mavisakalyan, Srinivas Goli.

**Writing – review & editing:** Anu Rammohan, Astghik Mavisakalyan, Srinivas Goli.

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
