## [Editor Report · Decision Letter 0]

18 Feb 2021

Dear Dr Goli, 

Thank you for submitting your manuscript entitled "Exposure to conflicts and its influence on continuum of maternal health care: Analyses of 452,192 women across 49 countries and 82 surveys" for consideration by PLOS Medicine for our upcoming Special Issue.

Your manuscript has now been evaluated by the PLOS Medicine editorial staff as well as by the Guest Editors, and I am writing to let you know that we would like to send your submission out for external assessment.

Once your full submission is complete, your paper will undergo a series of checks in preparation for external assessment. 

Kind regards,

Richard Turner, PhD

rturner@plos.org

---

## [Decision Letter · Decision Letter 1]

1 Apr 2021

Dear Dr. Goli,

Thank you very much for submitting your manuscript "Exposure to conflicts and its influence on continuum of maternal health care: Analyses of 452,192 women across 49 countries and 82 surveys" (PMEDICINE-D-21-00507R1) for consideration at PLOS Medicine for our upcoming Special Issue. 

Your paper was evaluated by the Guest Editors and sent to independent reviewers, including a statistical reviewer. The reviews are appended at the bottom of this email and any accompanying reviewer attachments can be seen via the link below:

[LINK]

In light of these reviews, we will not be able to accept the manuscript for publication in the journal in its current form, but we would like to invite you to submit a revised version that addresses the reviewers' and editors' comments fully. You will appreciate that we cannot make a decision about publication until we have seen the revised manuscript and your response, and we expect to seek re-review by one or more of the reviewers. 

We hope to receive your revised manuscript by April 26. Please email us (plosmedicine@plos.org) if you have any questions or concerns.

Please let me know if you have any questions, and we look forward to receiving your revised manuscript. 

Sincerely,

Richard Turner, PhD

rturner@plos.org

Please adapt the title to include a study descriptor after the colon, e.g., "...: A cohort study". 

Please add a new final sentence to the "Methods and findings" subsection of your abstract, beginning "Study limitations include ..." and quoting 2-3 of the study's main limitations. 

Given the study design, please avoid language implying causality, e.g., "effect" late in your abstract.

Please use the active voice in 1-2 points in your Author Summary (e.g., "We combined ..."). 

Please trim the final paragraph of the Introduction substantially. A brief description of the study's aim will suffice here, and discussion of the methods used and their merits can be moved to the Discussion section.

Early in the Methods section, please state whether or not the study had a protocol or prospective analysis plan, and if so attach the relevant document(s) as a supplementary file(s), referred to in the text. Please highlight analyses that were not prespecified. 

Please avoid claims of "the first" and the like, e.g., early in your Discussion section. Where necessary, please add "to our knowledge" or similar. 

Throughout the paper, please remove spaces from the reference call-outs (e.g., "... care usage [4,11,12], ").

Please review your reference list to ensure that references are formatted consistently and according to journal style. No more than 6 author names should be listed, followed by "et al.". Italics should be converted to plain text. Please abbreviate journal names consistently (e.g., "Lancet").

Please add a completed checklist for the most appropriate reporting checklist, e.g., RECORD, as a supplementary file, labelled "S1_RECORD_Checklist" or similar and referred to as such in your Methods section. In the checklist, please refer to individual items by section (e.g., "Methods") and paragraph number rather than by line or page numbers, as the latter generally change in the event of publication. 

Comments from the reviewers:

*** Reviewer #1: 

I confine my remarks to statistical aspects of this paper. Unfortunately, there were some fairly serious issues that need to be addressed.

One is how the fact that the data are nested was dealt with. There are a variety of approaches such as GEE, MLM and robust standard errors. I think a nonlinear MLM (multilevel model) is indicated here, but the other methods would also be OK.

Another is that the equation shown on p. 9 is not logistic regression, it is linear regression. You need to either change the DV by taking the logit or else exponiiante the right side.

Finally, causal language should be avoided; even with controls, this is an observational study.

More specific issues:

page 8 Don't make terciles or otherwise categorize continuous variables. This increases both type 1 and type 2 error and introduces "magical thinking" - that is, that something special happens at the cutoffs.

Talking about the "mean" of a dichotomous variable (like each stage of CoC) is a bit odd. Usually people would say "proportion" - but this works out the same.

The figures seem to have errors. By design, each stage of CoC has to be less common than earlier ones. But this is not reflected in the figure. E.g. for Afghanistan, the mean for level 3 is higher than for level 2. For Albania, they seem to be equal. Problems like this occur for other countries, too. So, either something is wrong in what you plotted, or I am confused (but if I am confused, others are likely to be confused as well).

*** Reviewer #2: 

The article "Exposure to conflicts and its influence on continuum of maternal health care: Analyses of 452,192 women across 49 countries and 82 surveys" by Rammohan explores the change in the odds of receiving any of a set of maternal care components in relation to exposure to conflict. The article is original, well-conceived, well-written, and compelling. 

The only substantive comment of note is that the exposure is defined at the provincial level. Some provinces (by which I assume the authors mean the first administrative level - the terminology differs by country) are small (and therefore exposure is well-measured), while others are large, and exposure may be measured with noise. Along a similar vein, using the province boundary may miss nearby conflicts just on the other side of the province boundary. An improvement would be to allow linkage by lat-long coordinates. However, that will likely only improve precision (classical measurement error in the exposure variables), and is not critical for inference in this paper.

The main non-substantive comments is that the manuscript is written as if geared for economists, and towing closer to medical literature templates may make it more impactful among medical/health readership. Suggestions may be to shorten/tighten the intro; remove boldface parts; simplify some of the methods; extend discussion and include some limitations.

The article is clear, lucid, and a pleasure to read. Overall this makes a really nice and important contribution to the thickening literature about the role of conflict in eroding the health and well-being of women and children.

*** Reviewer #3: 

Background: This section is rather thin and lacks the depth of the rationale for the study. It is not a surprise that conflict affects access to services. There is a lot of literature on this and it would have been useful to include even if small framework in what way the conflict affects this. For example evidence on increase of c-sections and delay of immunizations are key to understand the interaction between conflict and maternal health outcomes. Another consideration missing is type of place of conflict and how differently this might affect the access. Recent papers on Palestine (Leone et al and Siam et al.) have highlighted this mainly given different intensities (tackled by this paper) in different parts of the country. Basically the paper in the background and the conclusions needs to answer the so what question beyond the data contribution.

Methods: The key contribution of this paper is the inclusion of a large number of datasets and most importantly the inclusion of information of the Upsala datasets.

There are a few points that need clarification such as how were the data pooled together? Were synthetic cohorts used? Can you include more information on how the exposure was considered. Were past births included? Did you reconstruct the birth histories? How was the file constructed.

There needs to be a rationale for the outcomes. Eg: why 4 visits-not obvious to everyone. There needs to be a discussion on the quality of the conflict data, in particular how it varies across countries. This needs to go in the limitations.

Why was education not included? Was wealth calculated separately for urban and rural areas to account for different weight? If not it needs to be acknowledged in the limitations. 

Conclusions: this section is rather limited and needs more substance. It reiterates known messages and lacks depth. The limitations need to be thought through more as well as the implications.

***

[LINK]

---

## [Decision Letter · Decision Letter 2]

23 May 2021

Dear Dr. Goli,

Thank you very much for re-submitting your manuscript "Exposure to conflicts and its influence on continuum of maternal health care: Analyses of pooled cross-sectional data of 452,192 women across 49 countries and 82 surveys" (PMEDICINE-D-21-00507R2) for consideration at PLOS Medicine.

I have discussed the paper with editorial colleagues and the guest editors for the special issue, and it was also seen again by our reviewers. I am pleased to tell you that, provided the remaining editorial and production issues are dealt with, we expect to be able to accept the paper for publication in the journal.

[LINK]

Please let me know if you have any questions, and we look forward to receiving the revised manuscript.   

Sincerely,

Richard Turner, PhD

rturner@plos.org

Requests from Editors:

To your data statement, please add details of how UCPD data can be obtained. 

We ask you to adapt the title slightly to "Exposure to conflicts and the continuum of maternal health care: analysis of pooled cross-sectional data for 452,192 women across 49 countries and 82 surveys".

Please use tenses consistently in your abstract, e.g., "We observed ...". 

Prior to "We observed ..." in your abstract, we suggest adding a sentence to convey some summary statistics, e.g., the proportion of the sample exposed to violent conflict and/or the proportions accessing at least one antenatal service (with 95% CI) in conflict and non-conflict situations.

Where you quote an OR in the abstract, please include 95% CI and p values (e.g., "0.86, 95% CI 0.82-0.91, p<0.001").

Please adapt the abstract so that the final, single sentence of the "Methods and findings" subsection begins "Study limitations include ..." or similar and quotes 2-3 of the study's main limitations. The comment about "The UCDP used ... is generally considered to be reliable." should be removed or relocated to another part of the ms.

Late in the abstract, please make that "... statistically significantly and negatively associated ...".

Please state explicitly in the Methods section that the study did not have a prespecified analysis plan, assuming this is the case. 

The first paragraph of the discussion should summarize the study's findings: one way to achieve this would be to amalgamate the first two current paragraphs of that section, perhaps, with some trimming.

Where claiming a "first", "our study is unique", for example, please add "to our knowledge".

Please use the style "4 or more ANC visits" in the text, although numbers should be spelt out at the start of sentences. 

Please remove "hereafter" - an abbreviation on its own will suffice; italics should not be used for emphasis.

Please remove spaces from within reference call-outs throughout the ms, e.g., "[15-24,30-37]".

Please use journal name abbreviations consistently in the reference list, e.g., "Lancet" and "BMJ".

Noting reference 7 and others, please list 6 author names, followed by "et al." where appropriate.

In the labels for figure 4, we suggest adapting the text to "Exposure to violence in at least 1 conflict".

In the figures, we suggest adding "Point estimates with 95% confidence interval are shown." or similar to the legends and removing "95% confidence interval" from the figure files themselves.

Please break the attached STROBE checklist out into a separate supplementary file, labelled "S1_STROBE_Checklist" or similar and referred to as such in the text.

Comments from Reviewers:

*** Reviewer #1: 

The authors have addressed my concerns and I now recommend publication.

Peter Flom

*** Reviewer #2: 

The revised article "Exposure to conflicts and its influence on continuum of maternal health care: Analyses of 452,192 women across 49 countries and 82 surveys" by Rammohan improves on the original article by making some of the exposition more accessible.

I should note that the responses to the statistical reviewer are largely excellent. The reviewer brought up a couple of comments that commonly reflect an epidemiological perspective. As a quantitative researcher who often straddles methods from epidemiology and economics, it is easy to see why the reviewer would have the concerns about MLM and about the logistic exposition. At the same time, the authors had implemented everything correctly, and the gap was in the language differences between the fields. The latent variable model notation is standard econometrics (and used to motivate the use of LDV models), but not in epi. The variance-covariance clustering of standard errors is conceptually different from MLM, but in this case the single-level model is appropriate.

As noted before, making the exposition heed closer to the standards of medical literature would improve both the understanding and impact of the analysis. The authors have improved that somewhat, but I would urge them to continue hacking at this margin throughout the process of publishing this article.

I otherwise have no further comments.

*** Reviewer #3: 

The authors have definitely taken into account the comments and the paper is much stronger as a result. I would say the only shortcoming at the moment are the conclusions. Currently they are rather general and more of an afterthought. There needs to be more action beyond the international aid. What about emergency services and more practical solutions. A simple addition of a paragraph would do.

***

[LINK]

---

## [Editor Report · Decision Letter 3]

7 Jun 2021

Dear Dr. Goli,

Thank you very much for re-submitting your manuscript "Exposure to conflicts and the continuum of maternal health care: Analyses of pooled cross-sectional data for 452,192 women across 49 countries and 82 surveys" (PMEDICINE-D-21-00507R3) for consideration at PLOS Medicine.

I have discussed the paper with editorial colleagues and we will need to ask you to address some remaining issues listed at the end of this email. 

Please let me know if you have any questions, and we look forward to receiving the revised manuscript by Jun 14 2021 11:59PM.   

Sincerely,

Richard Turner, PhD

rturner@plos.org

Requests from Editors:

Please remove the final sentence of the "background" subsection of the abstract: this is a sentence of discussion.

Early in the abstract you mention that "Nearly 40% of the women in the sample had been exposed to conflict.", and later on quote a figure of 39.6%. If these two numbers are equivalent, please consider removing the earlier sentence.

At the end of the "Methods and findings" subsection of the abstract, please adapt the text to: "... other types of conflict-related harms.".

At the start of the "Conclusions" subsection of the abstract, please make that "statistically significantly and negatively associated ...". 

Where the word "gender" is used, e.g., in the Methods section, please substitute "sex" if appropriate. 

In the first paragraph of the Discussion section of the main text, please make that "These findings hold even after controlling for all other ...".

Please remove the information on data availability from the end of the main text (this will appear in the article metadata via entries in the submission form).

Noting reference 2 and others, please list 6 author names rather than 5, followed where appropriate by "et al.".

Please remove the "and" from the author list for reference 4.

Please use the journal name abbreviation "Lancet" consistently.

***

---

## [Editor Report · Decision Letter 4]

9 Jun 2021

Dear Dr Goli, 

On behalf of my colleagues and the Guest Editors, I am pleased to inform you that we have agreed to publish your manuscript "Exposure to conflicts and the continuum of maternal health care: Analyses of pooled cross-sectional data for 452,192 women across 49 countries and 82 surveys" (PMEDICINE-D-21-00507R4) in PLOS Medicine.

Prior to final acceptance, we suggest also quoting the OR for 4 ANC visits + institutional delivery + PNC in the abstract, including the non-significant p value, immediately prior to the sentence "We showed that both the incidence of exposure to conflict as well as conflict intensity have profound negative implications ...".

PRESS

Sincerely, 

Richard Turner, PhD 

rturner@plos.org